# The Effect of Substrate Properties on Cellular Behavior and Nanoparticle Uptake in Human Fibroblasts and Epithelial Cells

**DOI:** 10.3390/nano14040342

**Published:** 2024-02-10

**Authors:** Mauro Sousa de Almeida, Aaron Lee, Fabian Itel, Katharina Maniura-Weber, Alke Petri-Fink, Barbara Rothen-Rutishauser

**Affiliations:** 1Adolphe Merkle Institute and National Center of Competence in Research Bio-Inspired Materials, University of Fribourg, Chemin des Verdiers 4, 1700 Fribourg, Switzerland; mauro.sousadealmeida@unifr.ch (M.S.d.A.); a.lee22@imperial.ac.uk (A.L.); alke.fink@unifr.ch (A.P.-F.); 2Department of Bioengineering, Imperial College London, South Kensington, London SW7 2BP, UK; 3Empa, Swiss Federal Laboratories for Materials Science and Technology, Laboratory for Biomimetic Membranes and Textiles, Lerchenfeldstrasse 5, 9014 St. Gallen, Switzerland; fabian.itel@empa.ch; 4Empa, Swiss Federal Laboratories for Materials Science and Technology, Laboratory for Biointerfaces, Lerchenfeldstrasse 5, 9014 St. Gallen, Switzerland; katharina.maniura@empa.ch; 5Department of Chemistry, University of Fribourg, Chemin du Musée 9, 1700 Fribourg, Switzerland

**Keywords:** nanoparticle uptake, extracellular matrix, mechanobiology, nanofibers, stiffness, fibroblasts, epithelial cells

## Abstract

The delivery of nanomedicines into cells holds enormous therapeutic potential; however little is known regarding how the extracellular matrix (ECM) can influence cell–nanoparticle (NP) interactions. Changes in ECM organization and composition occur in several pathophysiological states, including fibrosis and tumorigenesis, and may contribute to disease progression. We show that the physical characteristics of cellular substrates, that more closely resemble the ECM in vivo, can influence cell behavior and the subsequent uptake of NPs. Electrospinning was used to create two different substrates made of soft polyurethane (PU) with aligned and non-aligned nanofibers to recapitulate the ECM in two different states. To investigate the impact of cell–substrate interaction, A549 lung epithelial cells and MRC-5 lung fibroblasts were cultured on soft PU membranes with different alignments and compared against stiff tissue culture plastic (TCP)/glass. Both cell types could attach and grow on both PU membranes with no signs of cytotoxicity but with increased cytokine release compared with cells on the TCP. The uptake of silica NPs increased more than three-fold in fibroblasts but not in epithelial cells cultured on both membranes. This study demonstrates that cell–matrix interaction is substrate and cell-type dependent and highlights the importance of considering the ECM and tissue mechanical properties when designing NPs for effective cell targeting and treatment.

## 1. Introduction

Intensive research in nanomedicine has been conducted to develop new systems such as nanoparticles (NPs) to deliver therapeutic agents, including drugs, to treat various diseases such as cancer [1,2]. The primary focus of researchers has been on engineering the physicochemical properties (e.g., size, shape, and surface chemistry) of these NPs to improve treatment efficacy [3,4]. However, more in-depth in vitro research into how cell NP uptake differs in healthy and diseased states is required. The efficient delivery of NPs to the target cells depends on cell mechanical properties and the physical and biochemical properties of the surrounding environment that directs cell behavior [5].

Tissue culture plastic (TCP) and track-etched porous membranes are the most commonly used reference substrate materials to grow cells in vitro for cell–NP interaction studies [6,7]. These membranes are usually incorporated into cell culture inserts and placed in well plates, separating the bottom (basal) and top (apical) compartments and facilitating the creation of physiologically representative barrier models [8,9]. These reference substrate materials are typically made of polystyrene, polyethylene terephthalate, and polyester and have mechanical behaviors significantly dissimilar to tissues in situ. Importantly, cells are influenced by substrate properties such as mechanical stiffness and consequently may account for differences in nanoparticle uptake behavior between cells cultured in vitro and cells cultured in vivo [10].

Biocompatible cell substrates such as scaffolds support cell attachment and growth. They are designed to mimic the extracellular matrix (ECM) properties of human tissues and are typically made up of fibrous components that act as attachment and growth cues for cells [11]. The ECM is a dynamic, acellular structure that, along with the surrounding cells, forms the tissue microenvironment [12]. The biomechanical properties of the ECM in human tissues can change in pathological conditions such as fibrosis and cancer [13]. Remodeling of the ECM in disease is typically associated with increased fiber deposition and overall stiffness [14,15]. Altered matrix stiffness and force signaling within the microenvironment can dramatically affect critical cell functions such as migration, proliferation, and differentiation [12,16]. Crucially, matrix stiffness can drive cell mechanotransduction pathways and promote the fibroblast-to-myofibroblast transition and the differentiation of mesenchymal stem cells [17,18]. Furthermore, changes in the ECM reorganization and topography, including fiber alignment, have been shown to limit tumor invasion in breast cancer [19]. This knowledge should be considered when designing and developing new therapeutics.

Investigation of cell behavior on cell substrates that more closely resemble the tissue microenvironment is required to support and facilitate data translation from in vitro culture to in vivo applications. We evaluate the role of substrate stiffness and topography in cell phenotype, inflammation, and the cellular uptake of NPs. Polyurethane (PU) has been used for various biomedical applications (e.g., scaffolds) as it is biocompatible and presents good mechanical properties (e.g., strength and elasticity) [20]. Electrospinning was used to create two soft and biocompatible PU membranes composed of non-aligned (randomly oriented) and aligned nanofibers. The influence of both soft membranes on the morphology of lung epithelial cells (A549) and fibroblasts (MRC-5) was investigated and compared with the stiff reference substrate TCP/glass. The biocompatibility of PU membranes and their effects on critical processes such as the inflammation and internalization of 80 nm silica (SiO_2_) NPs were investigated.

## 2. Materials and Methods

### 2.1. Materials

Texin 985A is a polyether-based thermoplastic polyurethane (PU) obtained from Covestro, Leverkusen, Germany. N, N-dimethyl formamide (DMF, >99.8%), tetraethylorthosilicate (TEOS, >99%), (3-aminopropyl)triethoxysilane (APTES, 99%), dimethylsulphoxide (DMSO, >99%), hexamethyldisilazane (HMDS, 99.9%), bovine serum albumin (BSA; >98% heat shock fraction, protease-free, fatty acid-free, essentially globulin free, pH 7), sodium azide (99%), ethylenediaminetetraacetic acid (EDTA), paraformaldehyde (PFA), Triton-X, and 4′,6-diamidino-2-phenylindole (DAPI) were obtained from Sigma Aldrich, Darmstadt, Germany. Sulfo-Cyanine 5 (Cy5) NHS ester was bought from Lumiprobe, Germany. Absolute ethanol (>99.8%) was purchased from VWR, Dietikon, Switzerland, and ammonia solution (25%) was purchased from Merck, Buchs, Switzerland. Phalloidin-AlexaFluor 488 (Cat #A12379) and secondary antibody goat anti-mouse AF647 (Cat #A21235) were bought from Invitrogen, Thermo Fisher Scientific, Basel, Switzerland. Transforming growth factor beta (TGF-β, Cat #11343160) and tumor necrosis factor alpha (TNF-α, Cat #11343015) were acquired from Immunotools, Friesoythe, Germany. All cell culture materials were purchased from Gibco, Thermo Fisher Scientific, Basel, Switzerland, unless otherwise specified. This study made use of only Milli-Q water.

### 2.2. Electrospinning and Characterization of Polyurethane Fibers

An in-house electrospinning setup was used to fabricate PU electrospun fiber membranes. A 12.5% *w/v* PU solution in DMF was prepared by dissolving 2.5 g of Texin 985A in 20 mL of DMF for 3 days under magnetic stirring at RT. To fully dissolve the polymer, the temperature was raised to 60 °C for 2 h. The solution was transferred into a 3 mL syringe with a 21G blunt end needle and placed on a syringe pump (AL-1000, Aladdin, Sarasota, FL, World Precision Instrument, USA). The pump flow rate was set to 400 µL/h, and random electrospun fibers were collected on a rectangular plate covered with aluminum foil placed 20 cm from the needle. The electrospinning process was run for 120 min at a constant voltage of 9 kV. A rotating drum spun at 1500 rpm was used to generate aligned fiber membranes. Before performing additional experiments, the samples were allowed to dry for a week at room temperature. The morphological properties of the electrospun fibers were evaluated using an S-4800 scanning electron microscope (SEM, Hitachi High-Technologies, Schaumburg, IL, USA) after sputter coating the samples with 8 nm Au/Pd (EM ACE600, Leica Microsystems, Opfikon, Switzerland). ImageJ (National Institutes of Health, Bethesda, MD, USA) and the OrientationJ plugin were used to determine the average diameter and orientation of the fibers. Tensile tests were conducted on a TA DMA Q800 dynamic mechanical analyzer. The samples were cut into rectangular shapes with a width of 5.27 mm and a gauge length of f ~15 mm. Tests were carried out at RT with a strain rate of 100% min^−1^. The elastic region of the stress–strain curve was used to calculate the Young’s modulus. Tensile data are reported as the mean of three independent measurements, and all errors are reported as the standard deviation. Micrometer measurements (Caliper, Mitutoyo MDH) were performed to determine the thickness of the membranes. Fourier-transform infrared (FTIR) spectroscopy of random and aligned fibrous membranes was performed using a Perkin Elmer (USA) Spectrum 65 FTIR spectrometer from 600 to 4000 cm^−1^ with 25 accumulations per sample at a resolution of 4 cm^−1^.

### 2.3. Synthesis of SiO_2_-Cy5 Nanoparticles

SiO_2_ NPs were synthesized via a modified Stöber method, as previously reported [21,22,23]. To obtain SiO_2_ NPs with sizes of ~80 nm, 104 mL of ethanol and 3.9 mL of ammonia solution were added into a round-bottom flask and magnetically stirred for 30 min at 60 °C. Approximately 11 mL of TEOS was added into the solution and stirred for 2 min, followed by the addition of 100 μL of a mixture containing 1 mg of sulfo-Cy5 NHS ester and 1.5 μL of APTES in DMSO. The reaction was allowed to run for 4 h, after which the obtained NPs were purified by means of dialysis for two weeks, with the first day in ethanol and the following days in ultrapure water (18.2 MΩ). The final solution containing NPs was kept in the dark at 4 °C. The physical properties, including size and shape, of the SiO_2_-Cy5 NPs were evaluated using a transmission electron microscope (TEM, FEI Tecnai G2 Filter, Hillsboro, OR, US). ImageJ’s particle size analysis was used to determine the particle size distribution [24]. The colloidal stability of NPs in water and cell culture medium was determined at 24 h through dynamic light scattering (DLS) at 90° with a commercial goniometer instrument (3D LS Spectrometer, LS Instruments AG, Fribourg, Switzerland). At least 10 independent measurements were performed. The hydrodynamic diameter and zeta potential of Cy5-labeled particles dispersed in water were determined with a 90Plus particle size analyzer and ZetaPALS (Brookhaven Instruments Corp., Holtsville, NY, USA).

### 2.4. Cell Culture

Human alveolar epithelial type II cells (A549) and human lung fibroblasts (MRC-5) from the American Tissue Type Culture Collection (ATCC) were used in this study. The cells were grown at 37 °C in 5% CO_2_ and 95% humidity and passaged twice a week at 80–90% cell confluence. A549 cells were kept in Roswell Park Memorial Institute (RPMI)-1640 cell culture media supplemented with 10% fetal bovine serum (FBS), 2 mM L-Glutamine, 100 U/mL penicillin, and 100 μg/mL streptomycin. The supplemented solution of RPMI-1640 is further mentioned as complete RPMI (cRPMI). Human lung fibroblasts were grown in minimum essential medium (MEM) GlutaMAX supplemented with 100 U/mL penicillin, 100 μg/mL streptomycin, and 1 × non-essential amino acids.

### 2.5. Cell Experiments

In this study, the effect of fibrous membranes on cell–particle interactions was evaluated. Electrospun membranes were cut into discs with a scalpel and separated from the aluminum foil with the help of tweezers. Glass coverslips of 13 and 22 mm were used as support for the membranes. The extra edges were folded into the bottom of the coverslip to keep the membranes stable and flat. The coverslips were placed into 24- and 12-well tissue culture plates. Sterilization was performed by exposing the material to UV light for 20 min and adding 70 vol.% ethanol for 30 min. The membranes were washed three times with PBS and left in the incubator overnight before cell seeding. A549 (passage range 4–20) and MRC-5 cells (passage range 4–15) were seeded at 53,000 cells/cm^2^ for 72 h and 48 h, respectively, before exposure to 20 µg/mL of NPs for 24 h. TGF-β (10 ng/mL for A549 and 5 ng/mL for MRC-5) and TNF-α (1 µg/mL for A549 and 10 ng/mL for MRC-5) were added for 24 h. Metabolic activity, fluorescence imaging, and SEM assays were performed in 24-well plates, whereas 12-well plates were used for cytotoxicity, inflammatory response, and flow cytometry assays.

### 2.6. Cytotoxicity

According to the manufacturer’s protocol, a lactate dehydrogenase (LDH) assay was used to determine cytotoxicity caused by either the fibrous membranes or the NPs (Roche, Mannheim, Germany). Approximately 100 µL of cell supernatant and 100 µL of LDH reagent were pipetted onto a 96-well plate for each measurement, which were carried out in duplicate. 0.2 vol.% Triton X-100 was used as a positive control. A spectrophotometer (Benchmark microplate reader, BioRad, Cressier, Switzerland) was used to detect LDH release in the form of formazan at 490 nm with a reference wavelength of 630 nm.

### 2.7. Metabolic Activity

A WST-1 assay was used to determine the effects of the fibrous material on the metabolic activity of A549 and MRC-5 cells. After growing on the fibrous membranes, the cells were washed twice with PBS before being exposed to 100 µL of WST-1 solution diluted at 1:10 in supplemented cell culture medium for at least 30 min. The formed formazan dye was measured spectrophotometrically at 450 nm. Measurements were performed in triplicate.

### 2.8. Inflammatory Response

The release of pro-inflammatory cytokines (Interleukin (IL)-6 and IL-8) in the cell culture medium was determined using a DuoSet Enzyme-Linked Immunosorbent Assay Development Kit (ELISA, R&D Systems, Zug, Switzerland) and the manufacturer’s protocol. TNF-α was used as a positive control. Standards and samples were measured in triplicates and duplicates. The concentration of both inflammatory mediators was determined based on the standard curves. The final values were normalized against the total protein content. Calculations were performed by fitting the values in a four-parameter logistic (4PL) model using GraphPad Prism 9 software (GraphPad Software Inc., San Diego, CA, USA).

### 2.9. Flow Cytometry

The cells grown in the 12-well plates and exposed to particles were washed three times with PBS and detached with 100 µL of trypsin–ethylenediaminetetraacetic acid (EDTA) for 10 min. Approximately 900 µL of supplemented cell culture medium was added, followed by centrifugation for 5 min at 300× *g* at 4 °C. The cells were then stained for 5 min in 200 µL of PBS containing 0.2 µg/mL of DAPI. The cells were washed by centrifugation and resuspended in 300 µL of flow cytometry buffer (PBS with 1 *w/v*.% BSA, 0.1 vol.% sodium azide, and 1 mM EDTA at pH 7.4). Data were collected using a BD LSR FORTESSA (BD Biosciences, San Jose, CA, USA) equipped with a violet laser and a red laser and 450/50 and 670/30 bandpass filters. Data were analyzed with FlowJo v10.7.1.

### 2.10. Fluorescence Imaging

Cells cultured on the fibrous substrates were washed three times. A solution of 4% PFA in PBS was used to fix the cells for 15 min. Permeabilization with 0.2 vol.% Triton X-100 in PBS for 10 min was performed, followed by blocking with 1.0% *w/v* BSA and 0.1 vol.% Triton X-100 in PBS. Primary antibodies, 4 µg/mL of mouse anti-E-cadherin and 5 µg/mL of mouse anti-α-SMA, in blocking buffer were added for 2 h, followed by the addition of secondary antibody goat anti-mouse AF647 at 1:400 for 1 h. Cell nuclei and F-actin staining with 1 µg/mL DAPI and 13.2 µM Alexa Fluor 488^®^ phalloidin were performed in PBS for 30 min. All steps were performed at room temperature, with three rounds of PBS washing in between. Coverslips were mounted using Kaiser’s glycerol gelatin and stored at 4 °C in the dark. Images were captured using an inverted confocal laser scanning microscopy system (CLSM, Leica, Stellaris 5, Heerbrugg, Switzerland) outfitted with Power HyD S detectors, a Plan-Apochromat 63x/1.4 Oil CS2 objective (Leica, Heerbrugg, Switzerland), and LAS X software version 3.0 (Leica). Image acquisition was performed sequentially with a field of view of 184.70 μm × 184.70 μm and a pixel density of 1024 × 1024. Three different laser excitation wavelengths were used: 405 nm (DAPI), 488 nm (Alexa Fluor 488), and 633 nm (Alexa Fluor 647).

### 2.11. Scanning Electron Microscopy Analysis of Cells

The cellular interaction with the fibrous membranes was visualized using scanning electron microscopy (SEM) (TESCAN Mira 3 LM field emission, Czech Republic). The samples were fixed using a PBS solution containing 2.5 vol.% glutaraldehyde and 2 vol.% PFA for 1 h. Following three washes with water, the samples were dehydrated using a series of alcohol washes (20, 30, 50, 70, 80, and 100 vol.% ethanol) for 10 min each before being treated with a solution of HMDS in ethanol (1:1 volume ratio). Before imaging, the samples were left to dry for 3 days and sputter-coated with 3 nm of gold (Sputter Coater 108 Auto, Cressington Scientific Instruments, Watford, UK).

### 2.12. Statistical Analyses

The Shapiro–Wilk normality test was used to determine whether the data distribution was normal (GraphPad Prism). An unpaired *t*-test was used to compare two independent groups, while one-way ANOVA (with Dunnett’s post hoc test for multiple comparisons) was used to compare more than two groups with one variable.

## 3. Results

### 3.1. Properties of Electrospun Polyurethane Fibers

Membranes of randomly oriented and aligned nanofibers were produced through electrospinning of a thermoplastic polyurethane solution (PU, Texin 985) using DMF as the solvent. Optimization of the electrospinning parameters, including polymer concentration, voltage, and flow rate, were tested. The chosen electrospinning conditions allowed for the production of homogeneous fibers in the nanoscale range (Figure 1). The random fibers presented a mean fiber diameter of 892 ± 157 nm. In contrast, the aligned fibers showed a mean fiber diameter of 882 ± 165 nm. The alignment was confirmed by determining the orientation of the fibers, with a clear direction observable for the aligned fibers (Figure 1B). The mechanical properties of the fibrous mats were investigated using a tensile test (Appendix A). The linear region of the stress-strain curve was used to calculate Young’s moduli. Tensile tests were performed on membranes with random fibers and fibers aligned along (II) and perpendicular (T) to the strain direction. Membranes with fibers oriented parallel to the strain direction exhibited similar Young’s moduli to those with randomly oriented fibers. Conversely, fibers oriented perpendicularly to the strain direction revealed a Young’s modulus 4–5 times lower than that of the randomly oriented fibers. The thickness of both mats was close to 50 µm, with average thicknesses of 69 ± 23 and 48 ± 3 µm for randomly oriented and aligned nanofibers, respectively.

FTIR spectroscopy was used to confirm the chemical properties of both materials after electrospinning (Appendix A). Both spectra are identical, with an absorption band at ~3300 cm^−1^ corresponding to NH stretching and two sharp peaks at ~2900 cm^−1^ associated with −CH_2_ stretching [25]. The absorption band at ~1700 cm^−1^ is also linked to a C=O group, common in polyurethane.

### 3.2. Cell Interaction with Electrospun Membranes

The interaction between cells and the fibrous membranes, including cell attachment and morphology, was evaluated via CLSM (Figure 2). The results show that both cell types attach to the fiber membranes. A lower cell density can be observed for A549 cells on both membranes compared to the glass condition. A similar morphology for the A549 cells was observed between the different conditions. MRC-5 cells appear more elongated on the fibers, with more evidence on the aligned fibers. The same behavior can be observed in the SEM pictures (Figure 3). MRC-5 cells tend to attach and elongate in the direction of the fiber.

### 3.3. Cytotoxicity, Metabolic Activity, and Inflammatory Response

Cytotoxicity, metabolic activity, and inflammatory response were assayed to evaluate the biocompatibility of the produced mats. Cellular toxicity was assessed by measuring the amount of LDH released in the cell culture medium (Figure 4A,B). A compromised cell membrane leads to the release of intracellular LDH. There were no significant differences in LDH release between cells growing on the fibrous mats and tissue culture plastic (TCP). Similarly, no significant differences were observed in the metabolic activity of both cells (Figure 4C,D). Metabolically active cells convert the tetrazolium salt WST-1 into formazan. Cells growing on aligned and non-aligned PU fibers showed no significant difference in metabolic activity (WST-1 reduction) (Figure 4C,D).

In response to environmental stress, lung cells secrete inflammatory cytokines such as IL-6 and IL-8, which alert resident and circulating immune cells [26,27]. The release of both interleukins was measured using ELISA and normalized against the total protein content (Figure 5 and Appendix A). TNF-α, at a non-toxic concentration, was used as a positive control for inducing cytokine release (Appendix A). Compared to TCP, IL-6 secretion increased about two-fold in epithelial cells on both fibrous membranes (Figure 5A), although the differences were not statistically significant. Contrastingly, IL-8 secretion increased significantly by more than two-fold on both substrates (Figure 5B). Significant differences were also found in fibroblasts, with IL-6 release increasing ~1.5 times for cells grown on both mats (Figure 5C). IL-8 release from fibroblasts, on the other hand, was significantly increased (two-fold) on aligned fibers but not on randomly aligned fibers (Figure 5D).

### 3.4. Formation of Stress Fibers Containing Alpha-Smooth Muscle Actin

For MRC-5 cells, immunostaining of alpha-smooth muscle actin (α-SMA), a protein found in stress fibers, was performed (Figure 6). In healing tissues, fibroblasts develop a contractile phenotype defined by cytoskeleton reorganization and the de novo expression of α-SMA [28]. MRC-5 cells grown on glass revealed a weak signal for α-SMA. Nonetheless, the signal increased when TGF-β1 was added. Similar observations were denoted for fibroblasts growing on the different PU fibers, with cells displaying a high presence of α-SMA.

### 3.5. Cellular Uptake of Nanoparticles

The cell type and the NP characteristics are critical factors in cell–NP interaction [23]. Other parameters, such as cell substrate, can influence cell behavior and, subsequently, the interaction and uptake of NPs. To investigate the impact of the produced fibrous membranes on NP uptake, Cy5-labeled SiO_2_ NPs were synthesized. The main NP properties, such as morphology and size, were characterized via TEM (Appendix A). The produced SiO_2_ NPs present a spherical shape and an average diameter of 83 ± 9 nm (Appendix A). Hydrodynamic diameter and zeta potential measurements were performed in water (Appendix A). The particles presented an average hydrodynamic diameter of 99 nm with a low polydispersity index (PDI) of 0.04. In addition, zeta potential measurements revealed strongly negative values (−46 ± 3 mV).

The colloidal stability of NPs in cell culture medium was assessed via DLS after 24 h (Appendix A). The correlation function between particles in the cell culture medium and water did not change substantially. The absence of a significant increase in lag time and a flat baseline confirm the stability of NPs in the cell culture medium.

An LDH assay was performed to evaluate the cellular cytotoxicity of SiO_2_ NPs on the various substrates (Appendix A). There were no significant differences in cell membrane permeability, i.e., cytotoxicity, after NP exposure to A549 and MRC-5 cells.

Flow cytometry and CLSM imaging were used to examine the cellular uptake of SiO_2_ NPs on different substrates (i.e., TCP vs. fibrous membranes) (Figure 7). Lung epithelial cells and fibroblasts were exposed to SiO_2_ NPs for 24 h. There were no significant differences in the flow cytometry median fluorescence intensity (MFI) values for epithelial cells on the different substrates (Figure 7A). Internalization of NPs in epithelial cells was observed for all substrates (i.e., TCP and aligned and non-aligned fibers) to similar degrees (Figure 7B). In contrast, the MFI values for fibroblasts on both non-aligned and aligned fibers were significantly higher (three-fold increase) than on TCP (Figure 7C). Intracellular localization of NPs in fibroblasts upon uptake was confirmed in all substrates (Figure 7D).

## 4. Discussion

The electrospinning of PU into fibrous membranes was used to create substrates that could mimic features of the ECM in lungs under healthy (random) and diseased (aligned) conditions. The ECM in the lungs is primarily composed of collagen with high tensile strength and elastin with low tensile strength [29]. Alveolar–capillary basement membrane (BM) is a specialized ECM responsible for maintaining the integrity of epithelial–endothelial cell layers [30]. In pathological conditions such as idiopathic pulmonary fibrosis (IPF), there is a loss of integrity of the alveolar epithelium and downregulation of cell–cell adhesion molecules, resulting in a compromised BM [31,32]. This is accompanied by ECM remodeling, typified by increased production of ECM proteins (e.g., collagen), fiber density, and alignment [33,34]. Consequently, ECM alignment and stiffness changes can change cell behavior and phenotype. Herein, the influence of fiber alignment on cell behavior was investigated. Mean fiber diameters of 892 nm and 882 nm were obtained for randomly oriented fibers and aligned fibers (fibers along the strain direction), respectively, which are values close to collagen and elastin fibers in human pulmonary alveolar walls (between 952 nm to 1270 nm) [35]. Additionally, a membrane stiffness of 1.4 MPa was obtained for both fibers, which is slightly lower but comparable to alveolar BM stiffness (between 2 and 3 MPa) [30].

Lung epithelial cells (A549) and fibroblasts (MRC-5) were cultured on soft PU membranes and stiff TCP/glass with stiffnesses in the GPa range. Both cell types could attach and grow on either PU membrane with no signs of cytotoxicity. However, compared to TCP/glass, a lower number of cells can be observed on PU membranes (Figure 2), confirmed by a two-fold decrease in total protein content (Appendix A). This decrease in cell number is most likely related to a reduction in cell proliferation. Cell morphology changes, such as elongated cells, were observed in fibroblasts growing in PU aligned fibers. Experiments were also carried out on spin-coated flat PU substrates, but cells were unable to attach, and these substrates were not used further.

Although no significant morphological changes were observed in epithelial cells, the biochemical response associated with cytokine release was altered. High levels of IL-6 and IL-8 are present in the early stages of IPF [36]. IL-6 is a crucial regulator of inflammation and tissue repair, and chronic activation of IL-6 signaling can contribute to the development of fibrosis [26]. IL-8 acts as a chemoattractant for immune cells, plays an important role in cell proliferation, invasion, and epithelial–mesenchymal transition, and is a driver of fibrotic progression [27,37]. Epithelial cells released higher levels of IL-8 when cultured on soft PU membranes than when cultured on TCP. Despite this, fiber alignment did not trigger significant differences in cytokine release. This demonstrates that cytokine release is triggered on soft PU membranes regardless of fiber alignment. In contrast, IL-6 release from fibroblasts increased in cells on both membranes compared to TCP, while fibroblasts cultured on aligned fibers elicited a stronger response. Significant differences in IL-8 release were observed for fibroblasts cultured on aligned fibers but not on non-aligned fibers. This outcome suggests that both cells can sense the physical changes in the environment (i.e., soft PU membranes) and convert them into biochemical signals (i.e., cytokine release) through mechanotransduction. Furthermore, the major differences in fibroblast cell morphology were related to greater cytokine release in cells on aligned fibrous membranes. Nevertheless, the differences in cytokine release in both cells cannot be directly attributed to one specific substrate feature (i.e., surface chemistry, topography, or stiffness). Even though we cannot isolate a specific substrate physical attribute, we may directly compare aligned and non-aligned fibers and observe a difference in IL-8 secretion in fibroblasts.

Lung diseases such as IPF are characterized by ECM remodeling and epithelial damage leading to several pathophysiological disorders, including fibroblast activation and differentiation to myofibroblasts [31,38]. TGF-β is a biomolecule capable of inducing fibroblast-to-myofibroblast transition (FMT) in vitro [39,40]. TGF-β causes disruptions in the homeostatic microenvironment by promoting cell activation, migration, or invasion in IPF [41]. This profibrotic cytokine promotes the production of proinflammatory and fibrogenic cytokines and collagen and the formation of α-SMA stress fibers [28,39,42]. Stress fibers containing α-SMA were observed for fibroblasts grown on both fibrous membrane types. This protein is a strong marker of myofibroblast differentiation, implying that PU membranes can induce fibroblast–myofibroblast transition (FMT) similar to that observed upon TGF-β stimulation of fibroblasts. However, to confirm a myofibroblast phenotype, the expression of additional FMT markers such as β-cadherins and mature focal adhesions should be assessed [40]. Similar observations were made for cardiac fibroblasts growing on randomly aligned and parallel aligned collagen fibers [43]. Fiber organization and orientation were important in determining the number of cardiac fibroblasts positive for α-SMA stress fibers cultured on both collagen fibers, which was higher than on a flat substrate.

Despite significant efforts to understand mechanotransduction in cells cultivated in various cell substrates and consequent cellular responses such as cell adhesion and inflammation, only a few studies have investigated NP cellular uptake in more relevant pathophysiological contexts. It has been shown that substrate properties such as stiffness and topography can influence the cellular uptake of NPs [44,45,46]. Here, we have demonstrated that the uptake of SiO_2_ NPs by epithelial cells is not affected by the cell substrate (i.e., TCP or fibrous membranes). However, fibroblasts cultured on fibrous membranes rather than TCP could internalize more SiO_2_ NPs, but fiber alignment did not affect particle uptake. According to Yhee et al., primary human fibroblasts cultured on TCP coated with a collagen matrix instead of uncoated TCP were able to internalize more glycol chitosan NPs [47]. They also concluded that macropinocytosis was the primary mechanism involved in the internalization of NPs. It has also been shown that cancer-associated fibroblasts, including myofibroblasts, present an increased rate of macropinocytosis in comparison with normal fibroblasts [48]. Lung fibroblasts cultured on PU membranes shown morphological changes and an increased quantity of α-SMA stress fibers. Increased NP uptake may be associated with increased macropinocytosis activity, but more research is needed to confirm this claim. This finding strongly demonstrates that the microenvironment surrounding lung fibroblasts influences NP cellular delivery.

In this study, we found that cell–substrate interaction influences the cellular delivery of SiO_2_ NPs in a cell-type specific manner. Fiber orientation (aligned vs. non-aligned) did not affect particle internalization; however, as previously mentioned, it is unclear whether the increased uptake of SiO_2_ NPs observed in fibroblasts is primarily influenced by substrate surface chemistry, stiffness, or topography (i.e., nanofibers vs. flat TCP/glass). Despite the limitations of the study, we were able to show that fibroblasts were more sensitive to changes in the extracellular environment due to differences in cell morphology, cytokine release, and NP uptake. Additionally, we believe that PU fibrous membranes are a better substrate for studying lung cell–NP interactions than TCP/glass substrates because they have better physicomechanical properties (i.e., topography and stiffness). Nevertheless, the fabrication of fibrous membranes made of PU blending with ECM proteins (e.g., collagen and elastin), which present mechanical properties that more closely resemble the ECM in healthy vs. diseased lungs, should be investigated.

## 5. Conclusions

This study suggests that cell–matrix interaction is substrate and cell-type dependent with consequences for cell behavior and NP internalization. PU fibrous membranes with different physical properties (i.e., topography and stiffness) compared to standard TCP increased cytokine release in epithelial cells and fibroblasts. Fiber alignment had little effect on epithelial cells but had an effect on fibroblast behavior by orienting cell alignment and increasing IL-8 secretion. The number of stress fibers containing α-SMA and the uptake of SiO_2_ NPs increased in fibroblasts cultured on both membranes. However, the different substrates did not considerably affect the uptake of SiO_2_ NPs in epithelial cells. This study emphasizes the significance of considering the ECM and tissue mechanical properties when designing and optimizing NPs for effective disease treatment. To progress in this direction, appropriate cell models, priming biological compounds, and more realistic cell substrates should be considered to more closely mimic and build a healthy/diseased model.

## Figures and Tables

**Figure 1 nanomaterials-14-00342-f001:**
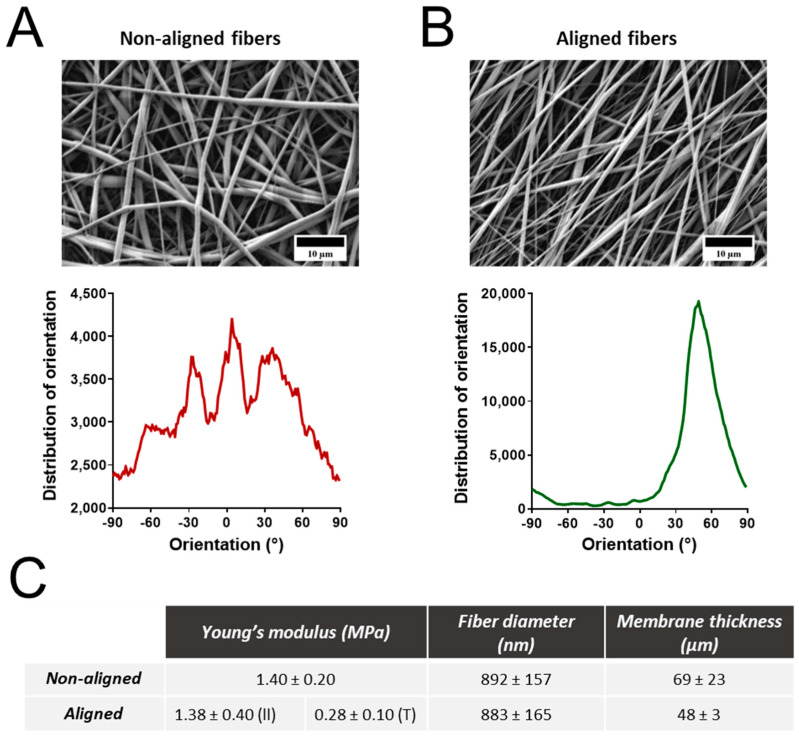
Characterization of non-aligned and aligned polyurethane electrospun fibrous membranes. Scanning electron microscopy images and fiber orientation of non-aligned (**A**) and aligned fibers (**B**). Tables presenting information regarding the Young’s modulus, average fiber diameter, and membrane thickness (**C**). Data are presented as the mean ± standard error of the mean. II—fibers along the strain direction; T—fibers perpendicular to the strain direction.

**Figure 2 nanomaterials-14-00342-f002:**
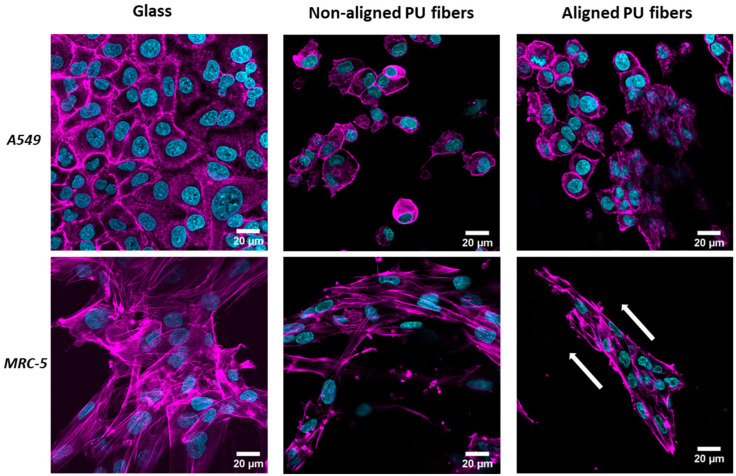
Cell interaction with different substrates. Confocal laser scanning microscopy (CLSM) images showing the morphology of the attached A549 lung epithelial cells (**top** row) and MRC-5 lung fibroblasts (**bottom** row) on glass, non-aligned polyurethane (PU) fibers, and aligned PU fibers. Nuclei are in cyan and f-actin is in magenta. Arrows indicate cell/fiber orientation.

**Figure 3 nanomaterials-14-00342-f003:**
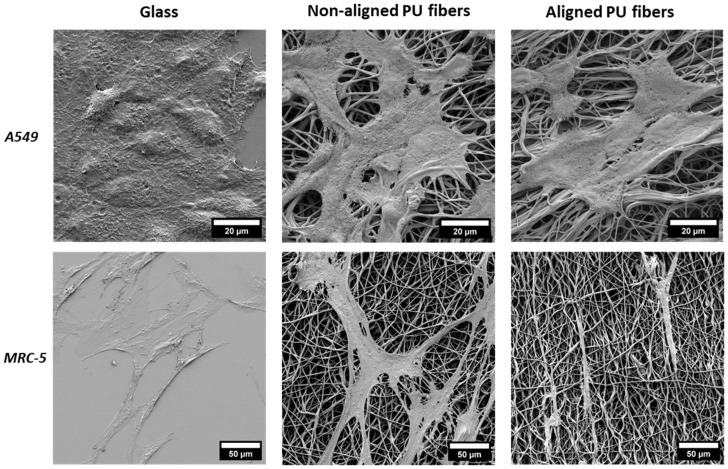
Scanning electron microscopy (SEM) images of lung epithelial cells (A549) and lung fibroblasts (MRC5) cultured on glass and aligned and non-aligned electrospun polyurethane (PU) membranes.

**Figure 4 nanomaterials-14-00342-f004:**
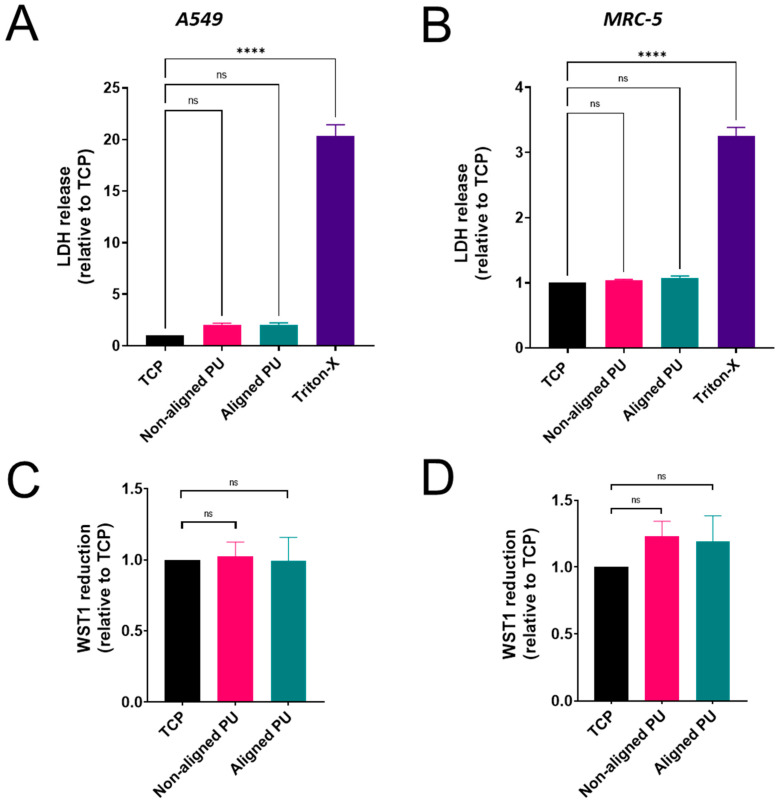
Impact of fibrous polyurethane (PU) membranes on the cytotoxicity and metabolic activity of lung epithelial cells (A549) and fibroblasts (MRC-5). Bar graphs, at the top, represent the release of lactate dehydrogenase relative to tissue culture plastic (TCP) from A549 (**A**) and MRC-5 (**B**) cells. Bar graphs, at the bottom, represent the reduction in WST-1 relative to TCP from A549 (**C**) and MRC-5 (**D**) cells. Absorbance values for the WST-1 assay were normalized against the total protein content. Data are presented as the mean ± standard error of the mean (*n* = 3). Statistical significance was determined via one-way ANOVA and Dunnett’s post hoc test for multiple comparisons and is represented by **** *p* ≤ 0.0001. ns: not significant.

**Figure 5 nanomaterials-14-00342-f005:**
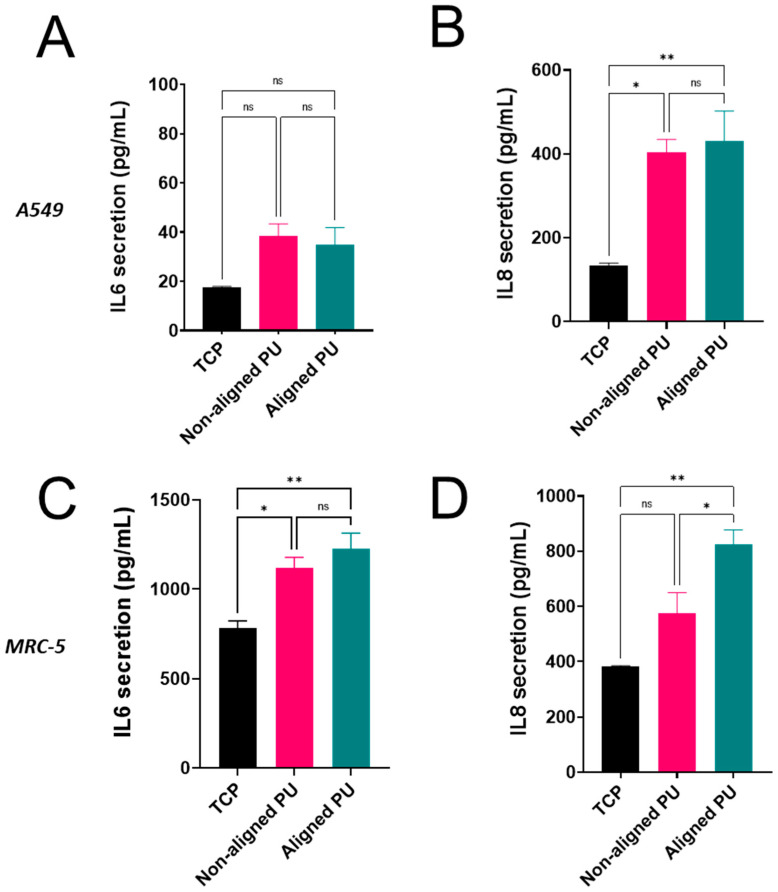
Impact of fibrous polyurethane (PU) membranes on the inflammatory response of lung epithelial cells (A549) and fibroblasts (MRC-5). Bar graphs represent the release of interleukin (IL)-6 and IL-8 from A549 (**A**,**B**) and MRC-5 (**C**,**D**) cultured on tissue culture plastic (TCP) and non-aligned and aligned PU. Values were normalized against the total protein content. Data are presented as the mean ± standard error of the mean (*n* = 3). Statistical significance was determined via one-way ANOVA and Tukey’s post hoc test for multiple comparisons and is represented by * *p* ≤ 0.05 and ** *p* ≤ 0.01. ns: not significant.

**Figure 6 nanomaterials-14-00342-f006:**
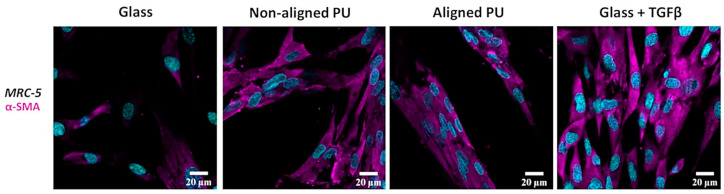
Immunofluorescence staining for α-smooth muscle actin (α-SMA). Confocal laser scanning microscopy (CLSM) images showing the immunostaining of MRC-5 fibroblasts with α-SMA cultured on glass, non-aligned and aligned polyurethane (PU) fibers, and glass + transforming growth factor beta (TGF-β). TGF-β was added for 24 h to MRC-5 cells at a concentration of 5 ng/mL. Nuclei are in cyan and α-SMA is in magenta.

**Figure 7 nanomaterials-14-00342-f007:**
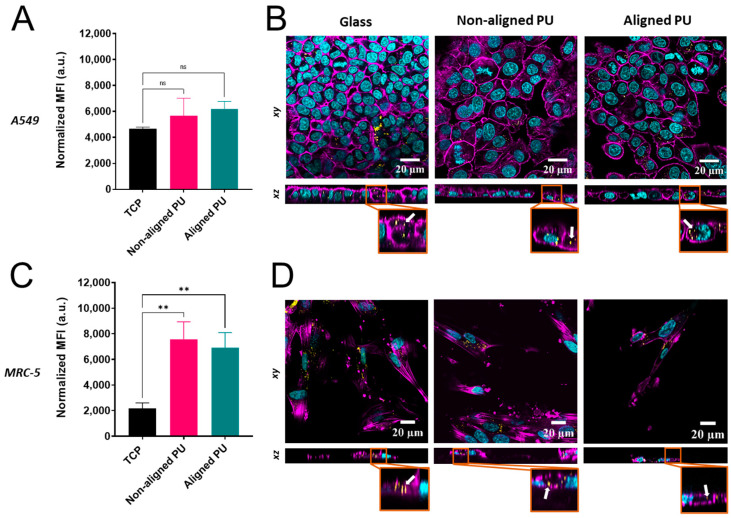
Impact of fibrous polyurethane (PU) membranes on SiO_2_ nanoparticle (NP) uptake in lung epithelial cells (A549) and fibroblasts (MRC-5). Bar graphs show the median fluorescence intensity (MFI) values obtained from flow cytometry measurements upon 24 h of exposure of 20 µg/mL SiO_2_ NPs to A549 (**A**) and MRC-5 cells (**C**) cultured on tissue culture plastic (TCP) and non-aligned and aligned PU fibers. Data are presented as the mean ± standard error of the mean (*n* = 3). Statistical significance was determined via one-way ANOVA and Dunnett’s post hoc test for multiple comparisons and is represented by ** *p* ≤ 0.01. ns: not significant. Confocal laser scanning microscopy (CLSM) images showing the internalization of SiO_2_ NPs in A549 (**B**) and MRC-5 cells (**D**) cultured on glass and non-aligned and aligned PU fibers. Nuclei are in cyan, F-actin is in magenta, and SiO_2_ NPs are in yellow. Arrows indicate the intracellular localization of NPs.

## Data Availability

The data presented in this study are openly available in the online repository Zenodo under the doi 10.5281/zenodo.10631246.

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
