# Peer review of "The Effect of Substrate Properties on Cellular Behavior and Nanoparticle Uptake in Human Fibroblasts and Epithelial Cells"

_nanomaterials, 2024, doi:10.3390/nano14040342_

Round 1

Reviewer 1 Report

Comments and Suggestions for Authors

Based on my experience of reviewing dozens of manuscripts for Nanomaterials and MDPI, this work is at a top level. It attempts to solve a very important basic scientific problem, that is, cell culture. The experimental design and structure of this work are reasonable, and the data is beautiful and of high quality. It can even be said that it has laid a certain foundation for the industrial production and application of petri dishes. But at present, there is a major problem that has not been solved, that is duplication. I hope the author can delve into this and solve it. In addition, there are some minor problems. If they can be solved, the quality of the manuscript will also be improved:

Major:

1.     The main issue is the orientation and reproducibility of aligned fibers, From Figure 1B and Figure S3, even the same materials (aligned fibers), the distribution of orientation is much different from batch to batch, which means the not good reproducibility, and which block its further applications in industry, so I’d suggest the authors to see this issue as a hard point and solve this problem first. If any strategy is used to solve this, please include the details and three microscopy images of three independent batches of aligned fibers with data of distribution of orientation in the revised manuscript.

2.     In figure 2, the cell density looks obviously different in different substates, does it mean the authors seed different number of cells or seed same number of cells, but the cells grow like that? Because the cell density also influences its growth and morphology, which is an important parameter for the experiments. So please make this point clear.

Minor:

1.     Italic for “in vivo”, “in vitro”, “in situ”

2.     Section 2.3, Line 135, it’s better to give more detailed process of using ImageJ to analyze size from TEM image or refer this paper to give readers guidance: Precise Analysis of Nanoparticle Size Distribution in TEM Image (Methods Protoc. 2023, 6(4), 63; https://doi.org/10.3390/mps6040063).

3.     For section 2.6, the volume of solution and type of slide and other parameters should be listed in LDH text.

4.     For section 2.10, please state the wavelength of 405nm, 488nm, and 633nm for which dye separately.

5.     Section 3.1, for FTIR analysis, it’s better to refer to the following papers: (1) Characterization of polyurethane resins by FTIR, TGA, and XRD. Journal of Applied Polymer Science, Volume 115, Issue 1 https://doi.org/10.1002/app.31096; (2) FTIR SPECTROSCOPY ANALYSIS OF THE PREPOLYMERIZATION OF PALM-BASED POLYURETHANE. Solid State Science and Technology, Vol. 18, No 2 (2010) 1-8.

6.     Line 333, Figure 2 should be Figure 5.

7.     In Figure 6B, in these two enlarged pictures, the cells pointed by the arrows have no nuclei. This is very strange. Can you explain it?

8.     “Scale bar = 20 µm” can be deleted from the caption in Figure 2, 5, and 6 since the scale bars shower in corresponding figures.

Reviewer 2 Report

Comments and Suggestions for Authors

The authors fabricated PU fibrous membranes with aligned and non-aligned nanofibers and investigated their impact on cell behavior and nanoparticle update in lung epithelial cells and lung fibroblasts. Although there are certain limitations of this study (e.g., cannot identify the exact features induce different cell behavior and nanoparticle uptake, lacking of ECM proteins to better mimic the real ECM structures), the authors have discussed and well justified those limitations in the manuscript. And the conclusion is well supported by the results. The manuscript can be accepted for publishing in Nanomaterials after minor revision.

Specific comments:

1. As the authors mentioned, the PU fibrous membranes affect the cell growth, so cell viability assay/proliferation assay should be conducted to further confirm the effect and biocompatibility of those two structures.

2. Figure 3 is mistakenly labelled as Figure 1.

3. Figure 5 is mistakenly labelled as Figure 2. The authors should be cautious of those mistakes, which make the manuscript hard to read and follow

4. The PU fibrous membranes can induce significant higher secretion of proinflammatory cytokines, especially for the MRC-5 cells. The authors may consider another control groups by adding PU into cell culture medium and culture cell with TCP, which can eliminate the effect generate from the PU itself.

5. Could the author explain why SiO2 NPs were selected to study the effect of PU fibrous membranes on cellular uptake? Several studies revealed that SiO2 NPs have toxicities, especially may not be suitable for in vivo applications or cell functional studies. A more biocompatible polymeric nanoparticles may be a better option.

6. Zeta potential results indicate that the SiO2 NPs are negatively charged. Is there any possible interaction between the SiO2 NPs with the PU fibrous membranes that may affect the cellular uptake? And the cell membrane charge may vary depends on the conditions, which may also affect the uptake of the surface negatively charged SiO2 NPs. The authors should clarify those unclear and unknown factors.

Reviewer 3 Report

Comments and Suggestions for Authors

This manuscript presents importance of physical characteristics of substrates that can influence cellular behaviour and subsequent nanoparticle uptake. It is an important factor to consider for nanomedicine delivery study. The manuscript presents extensive in vitro cell studies, and is well prepared. I have some comments as below:

1.       Page 6, line 261: Are the images from SEM or STEM? I was not sure if the word “Transmission” is a mistake. Please check.

2.       Page 8, line 294: it says “Figure 1” which should be “Figure 3”.

3.       Pages 8 and 9: Regarding data in these two figures, they are not normalized against cell numbers. As mentioned by authors and is clear from cell images, cell numbers on these substrates are quite different. Would you please describe how direct comparison was possible when these values are not normalized? 

4.       Page 10, line 333: “Figure 2” should be “Figure 5”.

5.       Regarding the choice of PU which might have desirable mechanical properties, chemistry of PU seems not supporting cell adhesion and growth effectively, unlike ECM. Attached cells may indeed be stressed which is not the case in vivo. I was wondering if you have studied attachment and proliferation of these two cell types on electrospun PU membranes and also considered about surface modification etc. to more closely resemble in vivo situation.

Reviewer 4 Report

Comments and Suggestions for Authors

Mauro Sousa de Almeida, Aaron Lee, Fabian Itel, Katharina Maniura-Weber, Alke Petri-Fink, Barbara Rothen-Rutishauser: Effect of substrate properties on cellular behavior and nanoparticle uptake in human fibroblasts and epithelial cells (# 2838058)

The topic of the manuscript is interesting. The works shows several novel findings, its methods, tests, assays are described precisely. However, the reader misses further information and details about the experiments.

Considering the Figure S3, the cells are on the outer surface of the fibrous matrix. Are the pictures typical shown in Fig. 2? Are the cells embedded between the fibres, even in partly?

Although, the parameter of the fibres in pulmonary system are mentioned, I suggest that the size of the voids (cages) is also important parameter. How does the mean distance, between the fibres, fit to the size of the cells? What can be occurred when the voids nearly the same (or larger) than the cells (in case of non-aligned and aligned PU fibres)?

I recommend inserting Fig. S3 into the manuscript from the supplementary matter. I suggest changing the magnification in Fig. S6 A.

Please correct the numbering of figures (second Fig.1 is Fig.3, second Fig. 2 is Fig. 5).

Round 2

Reviewer 3 Report

Comments and Suggestions for Authors

The authors have answered my comments and modified the data accordingly. I have no further comments. 

Reviewer 4 Report

Comments and Suggestions for Authors The manuscript has been sufficiently improved. I accept it for publication.